# Preclinical Efficacy of Peripheral Nerve Regeneration by Schwann Cell-like Cells Differentiated from Human Tonsil-Derived Mesenchymal Stem Cells in C22 Mice

**DOI:** 10.3390/biomedicines11123334

**Published:** 2023-12-17

**Authors:** Yu Hwa Nam, Saeyoung Park, Yoonji Yum, Soyeon Jeong, Hyo Eun Park, Ho Jin Kim, Jaeseung Lim, Byung-Ok Choi, Sung-Chul Jung

**Affiliations:** 1Department of Biochemistry, College of Medicine, Ewha Womans University, Seoul 07804, Republic of Korea; queennnam@ewha.ac.kr (Y.H.N.); saeyoung@ewha.ac.kr (S.P.); yjyum@ewha.ac.kr (Y.Y.); xoxo9540@ewhain.net (S.J.); 2Graduate Program in System Health Science and Engineering, Ewha Womans University, Seoul 07804, Republic of Korea; 3Cellatoz Therapeutics Inc., Seongnam-si 13487, Gyeonggi-do, Republic of Korea; hepark@cellatozrx.com (H.E.P.); hjkim@cellatozrx.com (H.J.K.); jlim@cellatozrx.com (J.L.); 4Department of Neurology, Samsung Medical Center, Sungkyunkwan University School of Medicine, Seoul 06351, Republic of Korea; bochoi@skku.edu

**Keywords:** Charcot–Marie–Tooth disease type 1A, tonsil-derived mesenchymal stem cells, Schwann cell-like cells, neuronal regeneration-promoting cells, C22 mice, peripheral nerve regeneration

## Abstract

Charcot–Marie–Tooth disease (CMT) is a hereditary disease with heterogeneous phenotypes and genetic causes. CMT type 1A (CMT1A) is a type of disease affecting the peripheral nerves and is caused by the duplication of the peripheral myelin protein 22 (*PMP22*) gene. Human tonsil-derived mesenchymal stem cells (TMSCs) are useful for stem cell therapy in various diseases and can be differentiated into Schwann cell-like cells (TMSC-SCs). We investigated the potential of TMSC-SCs called neuronal regeneration-promoting cells (NRPCs) for peripheral nerve and muscle regeneration in C22 mice, a model for CMT1A. We transplanted NRPCs manufactured in a good manufacturing practice facility into the bilateral thigh muscles of C22 mice and performed behavior and nerve conduction tests and histological and ultrastructural analyses. Significantly, the motor function was much improved, the ratio of myelinated axons was increased, and the G-ratio was reduced by the transplantation of NRPCs. The sciatic nerve and gastrocnemius muscle regeneration of C22 mice following the transplantation of NRPCs downregulated *PMP22* overexpression, which was observed in a dose-dependent manner. These results suggest that NRPCs are feasible for clinical research for the treatment of CMT1A patients. Research applying NRPCs to other peripheral nerve diseases is also needed.

## 1. Introduction

Charcot–Marie–Tooth (CMT) disease is one of the most common hereditary peripheral neuropathies. It is a rare disease with a worldwide prevalence of approximately 1 in 2500. The muscles in patients with CMT usually atrophy, and the shape of the hands and feet changes abnormally [1,2]. Over 40 genes have been validated in CMT, each of which is associated with one or more disease types. Additionally, one or multiple genes may be linked to one type of CMT [3]. CMT is classified as CMT type 1 (CMT1A), in which the myelin is damaged by genetic mutation; CMT type 2, in which the axon is damaged; CMT type X, with moderate nerve conduction; and CMT type 4, which is inherited as an autosomal recessive trait with impaired myelination [1,2,3]. The type that was previously named CMT type 3 is now classified as an infantile type (Dejerine–Sottas) or congenital hypomyelinated neuropathy [4].

The most common form, CMT1A, accounts for 60–70% of all CMT cases [1]. Patients with CMT1A are characterized by a reduced motor nerve conduction velocity (MNCV) < 38 m/s, reduced muscle stretching reflexes, and peripheral nerves with an “onion bulb” appearance. The disease occurs via the replication of a 1.4 Mb segment containing the gene for peripheral myelin protein 22 (*PMP22*) on chromosomes 17p11.2-p12 [5]. *PMP22* is mainly expressed in Schwann cells (SCs), constituting the myelin surrounding the axon of the peripheral nerves, and PMP22 is related to myelination during the development of the sciatic nerve. PMP22 is a quadrant helical integral membrane protein that is highly expressed as 2–5% of myelin protein weight in myelin-forming Schwann cells. The properly regulated expression and folding of PMP22 are essential for the development and maintenance of normal myelin in Schwann cells, and when this condition occurs abnormally, the overproduction of PMP22 proteins results in CMT1A [6]. Previously, the only methods for treatment were physical rehabilitation, pain control, and walking assistance, but now more than 100 related genes have been discovered, and the development of drug therapies [1,7,8,9] and gene therapies [2,10,11,12,13] is actively underway.

Mesenchymal stem cells (MSCs) have advantages as therapeutic agents, such as in proliferation, multipotency, immune regulation, and tissue regeneration. Due to continuous research on these MSCs, specific guidelines and quality control methods have been developed, and numerous clinical applications of MSCs are being attempted [14,15]. Compared to MSCs of other origins, tonsil-derived MSCs (TMSCs) have a relatively high yield and proliferation (i.e., shorter doubling times), so they are excellent for quantitative acquisition as a therapeutic agent. Tonsil tissue is readily obtained from a tonsillectomy, the most frequently performed minimally invasive surgery for patients aged 5 to 19 years [15,16]. TMSCs have the potential to differentiate into mesodermal lineages (bone, cartilage, fat, muscle, and tendon), endoderm (hepatocytes, PTH/insulin-releasing cells), and ectodermal lineages (neuron-like cells and glial or Schwann cell-like cells) [15,16,17,18,19,20].

SCs are essential for peripheral nerve development and regeneration and have an important function in myelinating peripheral nerve axons [21]. Unlike the central nervous system, the peripheral nervous system has a remarkable ability to regenerate after injury, and SC plasticity contributes significantly to this ability. SCs can attract damaged neurons by secreting neurotrophic factors such as nerve growth factor (NGF), brain-derived neurotrophic factor (BDNF), and glial cell-derived neurotrophic factor (GDNF) to aid in axon extension [22,23,24]. In particular, GDNF and BDNF were confirmed to have increased expression in SCs differentiated from TMSCs (TMSC-SC) in a previous study [20].

In the present study, we confirmed the therapeutic effect and peripheral nerve regeneration caused by transplanting TMSC-SCs named neuronal regeneration-promoting cells (NRPC) into C22 mice, a CMT1A disease model. The NRPCs were named after the characteristics of TMSC-SCs as Schwann cells and the characteristics of the neurotrophic effect of these cells in a previous study [25]. As a standardized index for peripheral nerve damage and functional recovery, the sciatic functional index (SFI) and nerve conduction study (NCS) were measured. Subsequently, the morphological observations of the sciatic nerve and gastrocnemius muscle were made using transmission electron microscopy (TEM) and immunohistochemistry (IHC). We performed the Rotarod test to evaluate motor function improvement, observed changes in *PMP22* expression in the sciatic nerve, and used fluorescence in situ hybridization (FISH) to confirm the presence of transplanted NRPCs in the gastrocnemius muscle 12 weeks after a repeated administration.

## 2. Materials and Methods

### 2.1. Cultivation of TMSCs and NRPCs

TMSCs were isolated and cultured, as previously described, from tonsils collected from patients who underwent tonsillectomy [16]. The study protocol was approved by the Ewha Womans University Medical Center institutional review board (IRB No. EUMC-2021-09-036). Informed written consent was obtained from all patients participating in the study. TMSCs were cultured in a 100 mm culture dish in high-glucose DMEM containing 10% fetal bovine serum (FBS; Invitrogen, Carlsbad, CA, USA) and 1% penicillin–streptomycin (Sigma-Aldrich, St. Louis, MO, USA). NRPCs were differentiated from TMSCs as previously described [20]. NRPCs were cultured in laminin-coated (Sigma-Aldrich) cell culture dishes containing DMEM/F12 supplemented with 10% FBS, 10 ng/mL of bFGF (PeproTech, London, UK), 5 ng/mL of platelet-derived growth factor-AA (PDGF, PeproTech), 14 μM of forskolin (Sigma-Aldrich), and 200 ng/mL of recombinant human heregulin-β1 (PeproTech). NRPCs were produced according to a standardized production process at the Cellatoz Therapeutic facility (Cellatoz Therapeutics Inc., Seongnam-si, Republic of Korea), and the characteristics of these cells are shown in Table 1. Cells for transplantation were stored in vials with a cryoprotectant (CryoStor^®^ CS10; CS10, BioLife Solutions, Bothell, WA, USA) and used immediately after thawing.

### 2.2. Animal Experiments

A diagrammatic summary of the experimental plan is provided in Appendix A.

#### 2.2.1. C22 Mice

The protocols used with the mice were approved by the institutional animal care and use committee (IACUC No. EUM20-053) of Ewha Womans University College of Medicine, and all experiments were performed in accordance with the approved guidelines and regulations. C22 male mice from the C57BL/6 black mouse background strain were purchased from Jackson Laboratories (Bar Harbor, ME, USA) and were used to establish a breeding colony at the Samsung Medical Center.

#### 2.2.2. Transplantation

Five-week-old male C22 mice were randomly allocated to 5 groups: 2 groups without NRPC transplantation (age-matched wild-type (W/T), CS10 only (sham)) and 3 groups with NRPC transplantation. The NRPC groups were injected intramuscularly into the thigh with 2.5 × 10^4^ (low), 2.5 × 10^5^ (medium), or 5 × 10^5^ (high) cells mixed in 100 μL of CS10 per leg. Transplantation efficacy tests were performed with single and repeated administrations of NRPCs, and the number of cells used in the test and the number of C22 mice per experimental group are described in Table 2. Repeated administration was performed once more 4 weeks after the first transplantation with the same dose as the single administration.

#### 2.2.3. Gait Test

After painting the soles of the hind legs of the mice with black ink and allowing them to walk about 50 cm on the paper, the footprint patterns were analyzed. This test was performed at 4-week intervals for 12 weeks and 16 weeks after single and repeated administration, respectively. The SFI was calculated by substituting the toe width and footprint length of each animal into the formula, and a quantitative analysis was performed with the value obtained [26].

#### 2.2.4. Rotarod Test

W/T and C22 mice were placed on a 3 cm horizontal rotating rod (12 rpm) after being pretrained for 7 days. A Rotarod machine with automatic timers and falling sensors (Jeung Do Bio & Plant Co., Seoul, Republic of Korea) was used. Every mouse was tested 3 times for up to 300 s each trial.

#### 2.2.5. Nerve Conduction Study

The mice were used for an electrophysiological study 12 weeks after the final transplantation. During the NCS, mice were anaesthetized with 3% isoflurane (Hana Pharm, Seoul, Republic of Korea) in an air mixture on a 37 °C heating plate. The NCS was performed using Nicolet VikingQuest apparatus (Natus Medical, San Carlos, CA, USA), and compound muscle action potential (CMAP) amplitudes and the MNCV were determined. The fur on the distal back and hind limbs was completely removed. Stimulating electrode spots were marked, and disposable subdermal needle electrodes (TE/S43-638, Technomed, Maastricht, The Netherlands) were used. The recording electrode was placed in the gastrocnemius muscle, and the reference electrode was placed in the Achilles tendon. Stimulating electrodes were placed on the biceps femoris to obtain a distal response and on the gluteus maximus to obtain a proximal response. The gap between the stimulating electrodes was 6 mm.

### 2.3. Transmission Electron Microscopy

The sciatic nerves were biopsied and fixed in 2.5% glutaraldehyde solution overnight. Semithin and ultrathin sections were prepared for ultrastructural observations. Sections were treated with 1% osmium tetroxide and embedded in epoxy resin. Semithin sections (0.5 μm) were stained with toluidine blue and observed using a microscope (BX51, Olympus, Tokyo, Japan). Ultrathin sections (70–80 nm) were counterstained with 0.25% lead citrate and 2% uranyl acetate and observed using a TEM (H-7650, Hitachi, Tokyo, Japan).

### 2.4. Immunohistochemistry

The mouse sciatic nerves and gastrocnemius muscles were fixed in 10% neutral buffered formalin (BBC Biochemical, Mount Vernon, WA, USA). After fixing for approximately 24 h at room temperature, these muscles were cut into 5 μm-thick serial sections and placed onto microscope slides. IHC was performed as previously described [20]. The stained tissues were photographed using a digital camera attached to a fluorescence microscope (Ti2-U, Nikon, Tokyo, Japan). The manufacturers and catalog numbers of antibodies used are provided in Appendix A.

### 2.5. Fluorescence In Situ Hybridization

The VividFISH FFPE pre-treatment kit (FP204) and FISH CEP probe for human chromosome Y-orange (FP117) (GeneCopoeia, Rockville, MD, USA) were used to detect human chromosome Y in the nerves of mice. The protocol was performed according to the instructions for the kit provided by the manufacturer. Briefly, tissue sections on slides were incubated in a paraffin pre-treatment solution at 85 °C for 90 min, washed in 2× saline sodium citrate, and incubated in a protease solution at 37 °C for 20 min. The sections were incubated in a denaturation solution at 73 °C for 5 min and treated with FISH probe mixed with the hybridization solution at 42 °C overnight. The slides were counter-stained with DAPI, and images were captured using a digital camera attached to a confocal fluorescence microscope (LSM800, Zeiss, Jena, Germany).

### 2.6. Real-Time Quantitative Polymerase Chain Reaction

MicroRNA isolation was carried out for cells using a mirVana miRNA Isolation Kit (AM1560, Invitrogen, Carlsbad, CA, USA). The isolated total RNA (10 ng) was reverse-transcribed to cDNA using the TaqMan Advanced miRNA cDNA Synthesis Kit (A28007, Applied Biosystems, Foster City, CA, USA) according to the manufacturer’s instructions. A polymerase chain reaction (PCR) was performed using TaqMan Fast Advanced Master Mix (4444556, Applied Biosystems) and TaqMan Advanced miRNA Assay (A25576, Applied Biosystems). The sequences of primer used were as follows: *hsa-miR-29a-3p*, 5′-UAGCACCAUCUGAAAUCGGUUA-3′ (478587_mir); *hsa-miR-221-3p*, 5′-AGCUACAUUGUCUGCUGGGUUUC-3′ (477981_mir); *hsa-miR-381-3p*, 5′-UAUACAAGGGCAAGCUCUCUGU-3′ (477816_mir). The sequence of *hsa-miR-423-5p* used as an endogenous control was 5′-UGAGGGGCAGAGAGCGAGACUUU-3′ (478090_mir). Real-time quantitative PCR (qPCR) reactions were performed by mixing 5 μL of diluted (1:10) cDNA template with 15 μL of reaction mixture containing 10 μL of TaqMan Fast Advanced Master Mix (2×), 1 μL of Taqman Advanced miRNA Assay (20×), and 4 μL of RNase-free water. The cycling conditions were as follows: 95 °C for 20 s, followed by 40 cycles of 95 °C for 3 s and 60 °C for 30 s. This was performed using the 7500 Fast RT-PCR system (Applied Biosystems, Thermo Fisher Scientific, Waltham, MA, USA). Relative gene expression was analyzed using the comparative Ct method (2^−ΔΔCt^). All measurements were carried out in triplicate.

### 2.7. Statistical Analysis

All experiments were performed at least 3 times. Statistical analyses were performed using Prism, version 9 (GraphPad Software, San Diego, CA, USA). The results are presented as mean ± standard error of the mean (SEM). The statistical significance of TMSCs and NRPCs was analyzed using Student’s *t* test. One-way analysis of variance (ANOVA) was used to determine any significant differences between the W/T, sham, and NRPC groups. A Bonferroni post hoc multiple comparisons test was used for the SFI and the grip test. A Tukey post hoc multiple comparisons test was used for the Rotarod test, NCS, and observations using TEM and IHC. Among ultrastructural analyses, “correlation between axon diameter and G-ratio” was analyzed using the simple linear regression method. A result of *p* < 0.05 was considered significant.

## 3. Results

### 3.1. NRPCs Improved Behavioral Test Results in a Single-Dose Transplantation Experiment

The NRPCs used for transplantation into C22 mice were differentiated from TMSCs. The MSC characteristics of TMSCs met the MSC criteria, such as differentiation potential, cell surface antigen, and cell shape, presented by the International Society for Cell Therapy (ISCT) (Appendix A). Also, the NRPCs used in this study exhibited morphological changes after differentiation, such as an elongated bipolar or tripolar spindle shape with a thinner and extended cytoplasm and larger nuclei than TMSCs, as observed in previous studies (Appendix A) [20].

After the single-dose transplantation, changes in the shape of the soles of the mice were observed through footprinting for gait testing (Figure 1A). The SFI according to these gait tests was calculated compared to W/T mice and showed a gradual recovery of motor function and was confirmed at 4, 8, and 12 weeks. Functional recovery was shown in the NRPC groups compared with the sham group, and recovery in the NRPC-high-dose group significantly improved at 8 and 12 weeks after transplantation in the right muscle. In the left muscle, there was a significant improvement in the NRPC-high-dose group at 8 weeks, and the NRPC-low-dose and -high-dose groups 12 weeks after transplantation (Figure 1B). NCS was performed 12 weeks after transplantation, which was the final time point of the behavioral experiment before tissue sampling. The NCS response was confirmed and quantified for each experimental group, and a representative graph is shown in Figure 1C. In the right leg, the nerve conduction velocity (NCV) tended to increase in the NRPC groups compared with the sham group, but the increase was not significant. The CMAP increased in the NRPC groups, especially in the NRPC-high-dose group. In the left leg, the NCV increased in the NRPC-high-dose group compared with the sham group, and the CMAP tended to improve in a dose-dependent manner in the NRPC groups, but the improvement was not significant (Figure 1D). The decrease in the NCV was associated with segmental demyelination and a loss of thick nerve fibers, and the increase in the CMAP suggests that the damaged motor nerves that control the muscles regenerated and that the corresponding damaged muscles were restored, at least to some extent.

### 3.2. Restoration of Sciatic Nerve Structure in a Single-Dose Transplantation Experiment

The sciatic nerve was observed and photographed at 3000× (Figure 2A) and 5000× (Figure 2B) magnifications using the TEM, and representative pictures are shown. The ratio and pattern of demyelinated nerve fibers seen in the C22 mice compared with the majority of myelinated nerve fibers in the W/T mice were observed for each experimental group. The G-ratio is obtained by dividing the diameter of an axon by the diameter of myelinated nerve fibers which inversely indicates the thickness of myelin and is also called the “axon/myelination ratio”. The C22 mice showed higher values 12 weeks after a single-dose transplantation than W/T mice. Compared with the sham group, the G-ratio in the NRPC-low-dose and -high-dose groups significantly decreased in the right leg, and it significantly decreased in the left leg in all the NRPC groups (Figure 2C). The ratio of myelinated axons to total axons was significantly increased in the NRPC groups compared with the sham group 12 weeks after transplantation in both the right and left legs (Figure 2D), which could predict remyelination by NRPCs.

### 3.3. Regeneration of the Sciatic Nerve and Gastrocnemius and the Related Pathway in a Single-Dose Transplantation Experiment

The sciatic nerves were collected 12 weeks after transplantation, and the myelination improvement was evaluated by the change in the expression of myelin basic protein (MBP) and the neurofilament heavy chain (NF-H) through IHC (Figure 3A). In the case of the W/T mice, the cross-section of myelin (MBP, green) surrounding the nerve (NF-H, red) appears round like a donut. In the enlarged picture, we observed a greater number of myelin-enclosed neurons in the NRPC groups than in the sham group, which indicates an increase in myelination by NRPC transplantation. Twelve weeks after transplantation, the gastrocnemius muscle was also collected and evaluated by IHC. The expression of myosin heavy chain 8 (MYH8), the most abundant protein in perinatal skeletal muscle, and laminin, a base substance that provides an anchor for the axons to skeletal muscle, were compared for each experiment group. The expression of MYH8 and laminin was increased in the NRPC groups, in which the staining pattern was similar to that in the W/T mice, and was the highest in the NRPC-high-dose group (Figure 3B). Moreover, the expression of myosin heavy chain 1E (MYH1E), the most abundant protein in adult skeletal muscle, was examined along with laminin. The staining confirmed that the expression of both proteins was greater in the NRPC groups than in the sham group (Figure 3C).

### 3.4. NRPC Improved Behavioral Test Results in the Repeated-Dose Transplantation Experiment

After the repeated-dose transplantation, the shapes of the soles of the mice were observed for SFI calculation as they were for single doses (Figure 4A). The function recovered continuously in the NRPC groups compared with that in the sham group and improved significantly in the NRPC-medium-dose and -high-dose groups by 8 and 12 weeks (on the right). On the left, a continuous recovery of function was also shown in the NRPC groups after transplantation compared with that in the sham group, and it improved significantly in the NRPC-medium-dose and -high-dose groups by 4, 8, and 12 weeks. In particular, there was a significant improvement in all the NRPC groups by 8 weeks (Figure 4B). When we measuring the NCS at 12 weeks after repeated-dose transplantation, a clear peak change was seen in the NRPC-high-dose group, although not as much as that in the W/T group (Figure 4C). The NCV of the right leg increased in proportion to the dose in the NRPC groups compared with the sham group, and the NCV in the NRPC-high-dose group was highly significant despite the large individual variation. The CMAP also increased in a dose-dependent manner in the NRPCs groups and was significantly higher in the NRPC-high-dose group (similar to the W/T mice) than in the sham or NRPC-low-dose groups. On the left, the NCV was significantly higher in the NRPC-high-dose group than in the sham group, and one mouse in this group showed a high NCV similar to that found in the W/T mice. There was a tendency for the CMAP to be improved dose-dependently in the NRPC groups compared with the sham group, but the improvement was only significant in the NRPC-high-dose group. The NCV in one mouse in the high-dose group was similar to the high NCV found in the W/T mice (Figure 4D). Moreover, 12 weeks after the repeated-dose transplantation, their motor ability was recovered, as confirmed by the Rotarod test. There was a continuous recovery of function in the NRPC groups compared with that in the sham group, and it was significantly improved in the NRPC-high-dose group (Figure 4E). Similar to the NCV results, one mouse in the NRPC-high-dose group had as long a latency on the rod as W/T mice.

### 3.5. Restoration of the Sciatic Nerve Structure in the Repeated-Dose Transplantation Experiment

The sciatic nerve was also observed using a TEM 12 weeks after the repeated-dose transplantation (Figure 5A,B). The G-ratio in the right leg of mice in the NRPC-medium-dose and -high-dose groups decreased significantly in a dose-dependent manner compared with the ratio in the sham group (Figure 5C). The G-ratio in the left leg of mice in the NRPC-medium-dose and -high-dose groups also decreased significantly in a dose-dependent manner compared with the ratio in the sham group. These results predict that remyelination will occur in the NRPC-medium-dose and -high-dose groups (Figure 5C). The ratio of myelinated axons to total axons was analyzed. On the right, the myelination in the NRPC-medium-dose and -high-dose groups increased significantly in a dose-dependent manner compared with that in the sham group. On the left, myelination in the NRPC groups increased significantly in a dose-dependent manner compared with that in the sham group (Figure 5D). An onion bulb appearance refers to the concentric collagen layer surrounding the SC and the axon, as seen in CMT1A, and hypertrophic neuropathy such as chronic inflammatory demyelinating polyneuropathy. On both the right and left sides, a significantly decreased onion bulb formation was only found in the NRPC-high-dose group, but it was seen in all NRPC groups in a dose-dependent manner compared with that in the sham group (Figure 5E).

The distribution of myelinated axon diameters was analyzed. The distribution in the right and left legs was almost identical. In particular, in the C22 mice, axons of myelinated nerve fibers were distributed evenly in the range of 10 μm or more, and the distribution of axons was rapidly reduced. As a result of classifying axons by dividing the axon diameter of the C22 mice into 2 μm sections, the number of axons in the repeated-dose NRPC groups was higher at all doses than that in the sham group. The NRPC-high-dose group had the most similar distribution to the W/T group, as 16–20 μm axons were consistently identified (Figure 5F). The correlation between the axon diameter and G-ratio showed a greater distribution in the NRPC-high-dose and the W/T groups at the 10–20 μm range and a lower G-ratio than in the sham, NRPC-low-dose, and NRPC-medium-dose groups. Ultimately, the distribution in the NRPC-high-dose group was most similar to the distribution in the W/T group (Figure 5G).

### 3.6. Restoration of the Sciatic Nerve and Gastrocnemius in the Repeated-Dose Transplantation Experiment

Twelve weeks after the repeated-dose transplantation, the sciatic nerves were examined using IHC. The enlarged image shows that the amount of myelin (MBP, green)-enclosed neurons (NF-H, red), such as those found in the W/T mice, increased in a dose-dependent manner in the NRPC groups. In particular, the size of the neurons surrounded by myelin in the NRPC-high-dose group was similar to that found in the W/T mice, which suggests that greater myelination is caused by repeated doses of NRPCs than single doses (Figure 6A). Furthermore, the expression of MYH8 and laminin was examined in the gastrocnemius muscle. Compared with the sham group, the expression was higher in the NRPC groups, and the muscle fiber morphology was similar to that seen in the W/T mice. Among the NRPC groups, the high-dose group showed the highest expression of MYH8 (Figure 6B). The expression levels of both proteins were higher in the NRPC-medium and high-dose groups than in the sham group, and the NRPC-high dose group showed similar staining patterns to the W/T group (Figure 6C).

### 3.7. Downregulation of PMP22 and the Presence of NRPCs 12 Weeks after Repeated-Dose Transplantation in the Sciatic Nerve

The downregulation of *PMP22* overexpression is the focus of therapeutic development for CMT1A. Twelve weeks after the repeated-dose transplantation, the expression of *PMP22* and myelin protein zero (*Mpz*) in the sciatic nerve was confirmed. When the expression of *PMP22* and that of *Mpz* were merged, the sham and W/T groups were dark orange and a pale orange close to yellow. The orange became lighter according to the administered transplantation dose, and the NRPC-high-dose mice showed a similar color to the W/T mice. When we normalized *PMP22* expression to the expression of *Mpz*, the relative expression of *PMP22* in W/T mice (1.095 ± 0.026) was close to 1.0 and similar to the expression found in the NRPC-high-dose group (1.17 ± 0.038). The expression in the NRPC-medium-dose (1.58 ± 0.069) and high-dose groups was significantly lower than it was in the sham group (1.9 ± 0.103) (Figure 7A).

To confirm whether the NRPCs were localized in the nerve 12 weeks after the repeated-dose transplantation, the sciatic nerves were examined using FISH. The NRPCs were differentiated from human TMSCs isolated from the tonsils of a 10-year-old boy. Therefore, we conducted FISH using a human chromosome Y probe and found a signal (red) for human chromosome Y in the epineurium of the C22 mice transplanted with NRPCs. In contrast, there was no signal in the mice from either the sham or W/T groups (Figure 7B).

### 3.8. Expression of Peripheral Nerve Regeneration-Related microRNA in NRPCs

To find the mechanisms of peripheral nerve regeneration with NRPC transplantation, a real-time qPCR was performed by extracting microRNA from TMSCs and NRPCs. The expressions of *miR-29a*, *miR-221*, and *miR-381* were normalized to that of miR-423 and compared relative to the TMSCs (Figure 8). As a result, the expression of all microRNAs (*miR-29a*: 3.87 ± 0.056; *miR-221*: 3.63 ± 0.014; *miR-381*: 3.52 ± 0.133) in NRPCs increased about fourfold compared to TMSCs (*miR-29a*: 1.00 ± 0.039; *miR-221*: 1.00 ± 0.041; *miR-381*: 1.00 ± 0.020). These three microRNAs have already been reported to be related to peripheral nerve regeneration and were used as mechanisms for verifying the efficacy of NRPCs in this study.

## 4. Discussion

Despite many attempts to develop CMT treatments, there are few specific treatments for this disease other than genetic counseling, symptomatic treatment, physical therapy, and rehabilitation treatments [27]. However, many treatments for CMT patients have still been studied, and drug therapies, including ascorbic acid, progesterone antagonists and modulators, PXT3003, HDAC6 inhibitors, P2X7 receptor modulators, and cytokines, have been reported [1,7,8,9]. Recently, gene therapies, including AAV-Neurotrophin-3, antisense oligonucleotides, siRNA, AAV9-shRNA, CRISPR/Cas9-sgRNA, and AAV9-miRNA, have also been studied [2,10,11,12,13]. Several stem cell therapies for CMT have also been conducted, but information on the development of cell therapy products remains limited. Unlike undifferentiated MSCs, which have been mainly used in attempts to develop cell therapy products, NRPCs are differentiated cells from TMSCs and are expected to be excellent candidates for CMT diseases.

The therapeutic strategy for CMT1A has focused on downregulating the overexpression of the *PMP22* gene. Therefore, various mouse models such as TgN248, JP18, JP18/JY13, trembler-J, C3, C61, My41, and C22 were created [28,29,30,31]. Neurotrophic factors secreted from the TMSC-SCs were able to affect the axon regeneration of damaged sciatic nerves in trembler-J mice [20]. However, the trembler-J mouse is a mild injury model caused by the *PMP22* point mutation and is mainly used as a representative model for CMT1E [29]. Therefore, we used the C22 mouse model, which is one of the most frequently used mouse models of CMT1A, with seven copies of human *PMP22*, including about 40 kb in the proximal region [32]. A C22 mouse shows symptoms of neuromuscular impairment due to distinct myelination abnormalities. The symptoms and signs, ultrastructural findings, the myelination pattern of peripheral nerves, and nerve conduction test results in C22 mice reflected those found in CMT1A patients. In patients with the CMT1A type, the more severe the disease, the more difficult it is to bend the instep, limiting bending and deforming the foot. These characteristics also appear in C22 mice, and the severity can be evaluated using the SFI by gait analysis [33].

We have previously shown that TMSCs can differentiate into SCs, which can serve as an alternative cell source for native SCs. TMSC-SCs showed myelination when co-cultured with mouse dorsal root ganglion (DRG) neurons and therapeutic effects when transplanted into mice with a sciatic nerve injury [25]. In a subsequent study, TMSC-SCs were transplanted into trembler-J mice, a model of CMT1A, and induced sciatic nerve and skeletal muscle regeneration [20]. These positive effects may result from neurotrophic factors, such as BDNF, GDNF, and NGF, produced by the SCs. In addition, TMSC-SCs have also been used in studies to test their potential for therapeutic treatment. After a co-culture of TMSC-SCs from healthy donors and CMT patients with DRG neurons, the effect of treatment was confirmed by myelination [8]. The TMSC-SCs (NRPCs) are expected to have the potential for CMT therapeutics.

Although a clear mechanism for peripheral nerve regeneration and muscle recovery after the NRPC transplantation was not confirmed in this study, several speculations were made that the NRPCs helped nerve regeneration. As the basis, first, NRPCs could infiltrate the epineurium, where they play the role of SCs. Twelve weeks after the repeated-dose transplantation, NRPCs were detected in the nerves of the C22 mice transplanted with NRPCs using FISH. These results can provide evidence that NRPCs entered the transplanted site and can be directly involved in regeneration. Second, it might be from the paracrine effect of NRPCs, secreting factors might stimulate peripheral nerve regeneration. In addition to differentiating into appropriate cells, stem cells secrete bioactive neurotrophic molecules to provide a microenvironment conducive to neuronal survival and neurogenesis [34]. Finally, SCs naturally express microRNAs that regulate gene expression, such as *miR-29a* and *miR-381*, which are regulators of *PMP22*. There have been studies that reduced PMP22 expression with *miR-381* in CMT model mice [35] and with *miR-29a* in CMT model cells [36]. It has also been demonstrated that *miR-29a* inhibits *PMP22* expression in SCs [37]. *miR-221-3p* promotes SC proliferation and migration and is used as a therapeutic strategy to promote nerve regeneration and functional recovery [38]. In this study, the increased expression of *miR-29a-3p*, *miR-221-3p*, and *miR-381-3p* in NRPCs compared to TMSCs was confirmed by qRT-PCR.

Here, we significantly improved gait in the NRPC groups compared with the sham group, suggesting that NRPCs support robust axon outgrowth and the structural formation of myelin. An improvement in remyelination after NRPC transplantation in a dose-dependent manner was confirmed by an increase in myelin thickness seen with a TEM, which was similar to that seen in the W/T mice. Compared with other studies [7,8,9,10,11,12,13], the present study clearly found a treatment effect on myelin sheath thickness and a correlation between the axon diameter and G-ratio. In addition, significant improvements in the NCV and CMAP were confirmed in the NRPC groups, which indicates that NRPCs promote axonal regrowth and remyelination. In a sciatic nerve-injured mouse, the nerve fibers or axons had to reach the muscles within 5 weeks to form synapses after regenerating [39], and the regeneration of peripheral nerves and adjacent muscles appears to be highly correlated. The thigh muscle near the nerve could be a suitable choice for the NRPC implantation site in this study.

## 5. Conclusions

NRPC efficacy evaluation tests, including behavioral observations, gait tests, nerve conduction tests, and morphological analyses based on immunohistochemical and ultrastructural findings in C22 mice, a mouse model for CMT1A, indicate both nerve and muscle regeneration in the NRPC groups. Most of these appeared in a dose-dependent manner. In particular, all the experimental results of the repeated-dose transplantation of the NRPC-high-dose group indicated superior recovery compared with all of the experimental results for the single administration of corresponding doses of NRPCs in the C22 mice. NRPCs will be useful as a treatment for CMT disease and will also be effective in other peripheral nerve treatments.

## Figures and Tables

**Figure 1 biomedicines-11-03334-f001:**
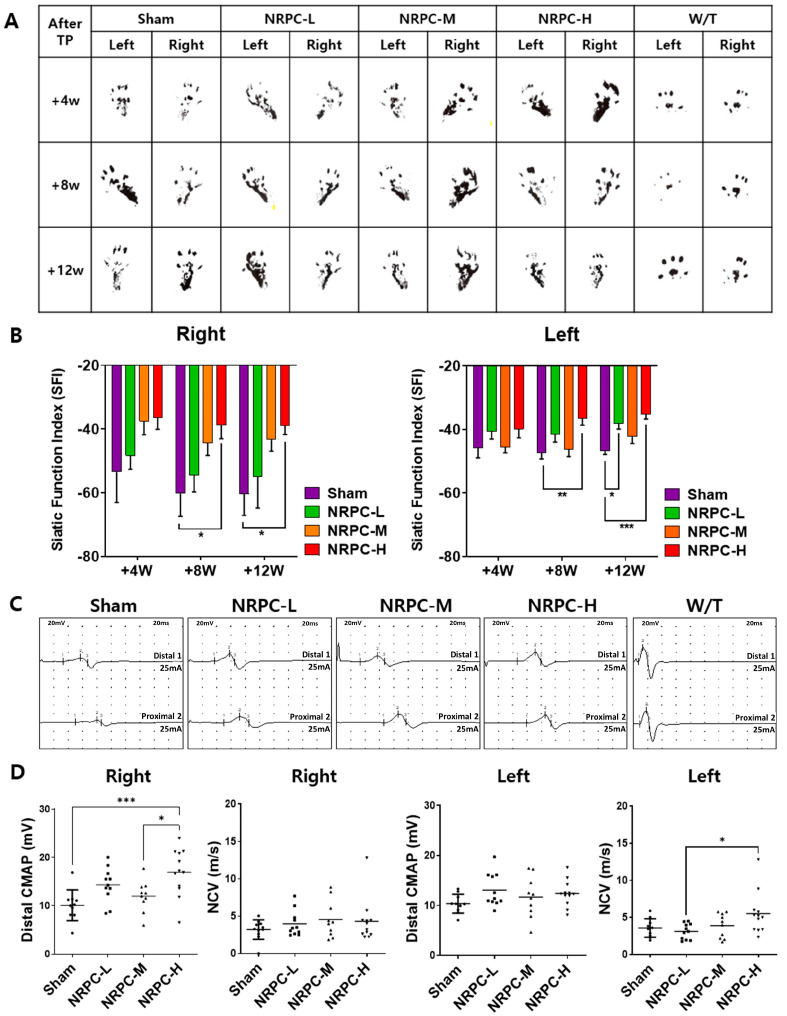
Improvement in motor function following single-dose transplantation with NRPCs using the gait test and NCS. Image of footprint (**A**) and graph of SFI (**B**) to confirm motor function recovery in gate test at 4, 8, and 12 weeks after transplantation. Representative graph of NCS (**C**) and CMAP and NCV measurements (**D**) 12 weeks after transplantation. Statistical analysis with one-way ANOVA was performed for comparison between groups, and data are presented as mean ± SEM (* *p* < 0.05; ** *p* < 0.01; *** *p* < 0.001). NRPCs, neuronal regeneration-promoting cells; NCS, nerve conduction study; SFI, sciatic functional index; CMAP, compound muscle action potential; NCV, nerve conduction velocity; NRPC-L, NRPC-low; NRPC-M, NRPC-medium; NRPC-H, NRPC-high; W/T, wild-type.

**Figure 2 biomedicines-11-03334-f002:**
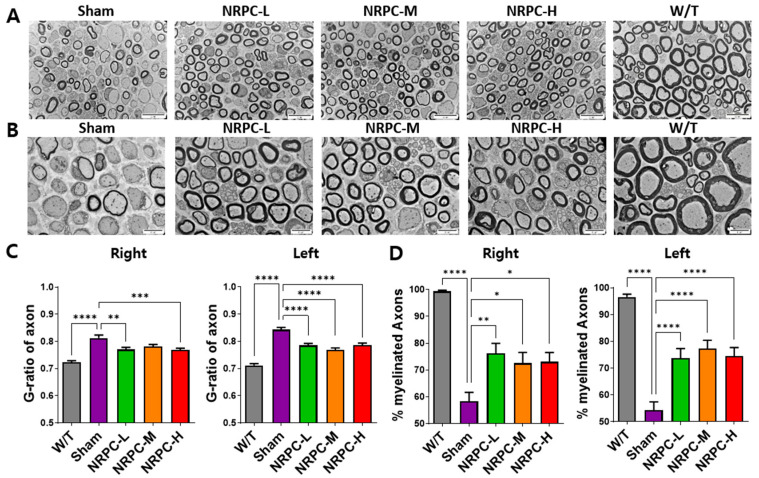
Myelination of the sciatic nerve in C22 mice single-dose transplanted with NRPCs using a transmission electron microscope. Representative images at 3000× (**A**) and 5000× (**B**) magnification of sciatic nerve cross-sections are presented. Scale bar of images at 3000×: 20 μm. Scale bar of the images at 5000×: 10 μm. Analysis of myelin thickness recovery (**C**) and ratio of myelinated nerve fibers (**D**) of the sciatic nerve after NRPC single-dose transplantation. Statistical analysis with one-way ANOVA was performed for comparison between groups, and data are presented as mean ± SEM (* *p* < 0.05; ** *p* < 0.01; *** *p* < 0.001; **** *p* < 0.0001). NRPCs, neuronal regeneration-promoting cells; NRPC-L, NRPC-low; NRPC-M, NRPC-medium; NRPC-H, NRPC-high; W/T, wild-type.

**Figure 3 biomedicines-11-03334-f003:**
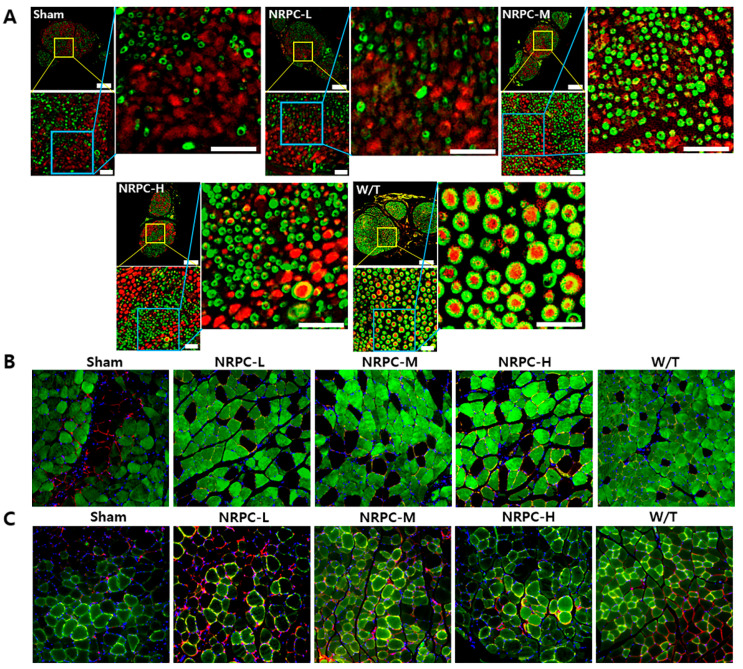
Observation of C22 mouse nerves and muscles using immunohistochemistry in single-dose transplantation with NRPCs. (**A**) Axon regeneration and myelin formation of sciatic nerves were confirmed by staining of NF-H (red) and MBP (green). Scale bar of original image: 100 μm. Scale bar of the enlarged yellow and blue box: 20 μm. (**B**,**C**) To confirm the gastrocnemius muscle reconstruction, MYH8 (green) (**B**) or MYH1E (green) (**C**) was double-stained with laminin (red), respectively (DAPI (blue)). Original magnification: 200×. NRPCs, neuronal regeneration-promoting cells; NF-H, neurofilament heavy; MBP, myelin basic protein; MYH8, myosin heavy chain 8; MYH1E, myosin heavy chain 1E; NRPC-L, NRPC-low; NRPC-M, NRPC-medium; NRPC-H, NRPC-high; W/T, wild-type.

**Figure 4 biomedicines-11-03334-f004:**
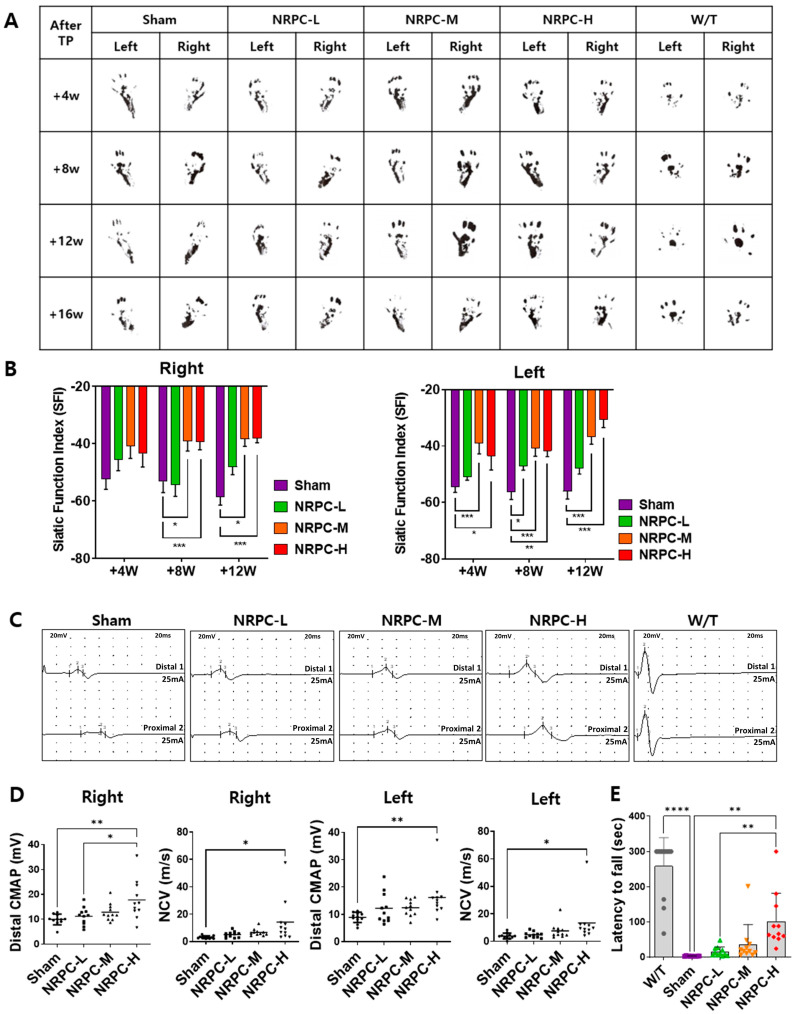
Improvement in motor function following repeated-dose transplantation with NRPCs using gait test, NCS, and Rotarod test. Image of footprint (**A**) and graph of SFI (**B**) to confirm motor function recovery in gate test 4, 8, and 12 weeks after transplantation. Representative graph of NCS (**C**) and CMAP and NCV measurements (**D**) 12 weeks after transplantation. Confirmation of motor function recovery in C22 mice with repeated-dose transplantation with NRPC by Rotarod test (**E**). Statistical analysis with one-way ANOVA was performed for comparisons between groups, and data are presented as mean ± SEM (* *p* < 0.05; ** *p* < 0.01; *** *p* < 0.001; **** *p* < 0.0001). NRPCs, neuronal regeneration-promoting cells; NCS, nerve conduction study; SFI, sciatic functional index; CMAP, compound muscle action potential; NCV, nerve conduction velocity; NRPC-L, NRPC-low; NRPC-M, NRPC-medium; NRPC-H, NRPC-high; W/T, wild-type.

**Figure 5 biomedicines-11-03334-f005:**
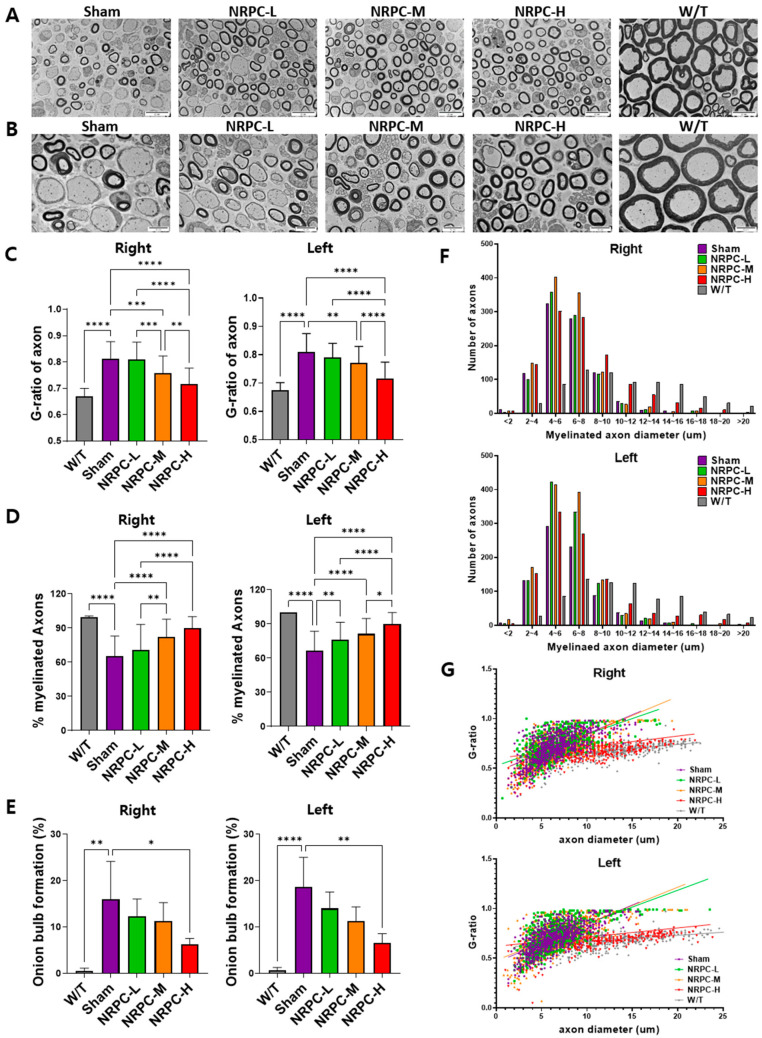
Myelination of the sciatic nerve in C22 mice with repeated-dose transplanted with NRPCs using transmission electron microscope. Representative images at 3000× (**A**) and 5000× (**B**) magnification of sciatic nerve cross-sections are presented. Scale bar of images at 3000×: 20 μm. Scale bar of the images at 5000×: 10 μm. Analysis of myelin thickness recovery (**C**) and ratio of myelinated nerve fibers (**D**) of the sciatic nerve after NRPC repeated-dose transplantation. Analysis of the ratio of onion bulb formation (**E**), the distribution according to diameter of myelinated axons (**F**), and the correlation between axon diameter and G-ratio (**G**) through NRPC repeated-dose transplantation in the sciatic nerve. Statistical analysis with one-way ANOVA was performed for comparison between groups, and data are presented as mean ± SEM (* *p* < 0.05; ** *p* < 0.01; *** *p* < 0.001; **** *p* < 0.0001). NRPCs, neuronal regeneration-promoting cells; NRPC-L, NRPC-low; NRPC-M, NRPC-medium; NRPC-H, NRPC-high; W/T, wild-type.

**Figure 6 biomedicines-11-03334-f006:**
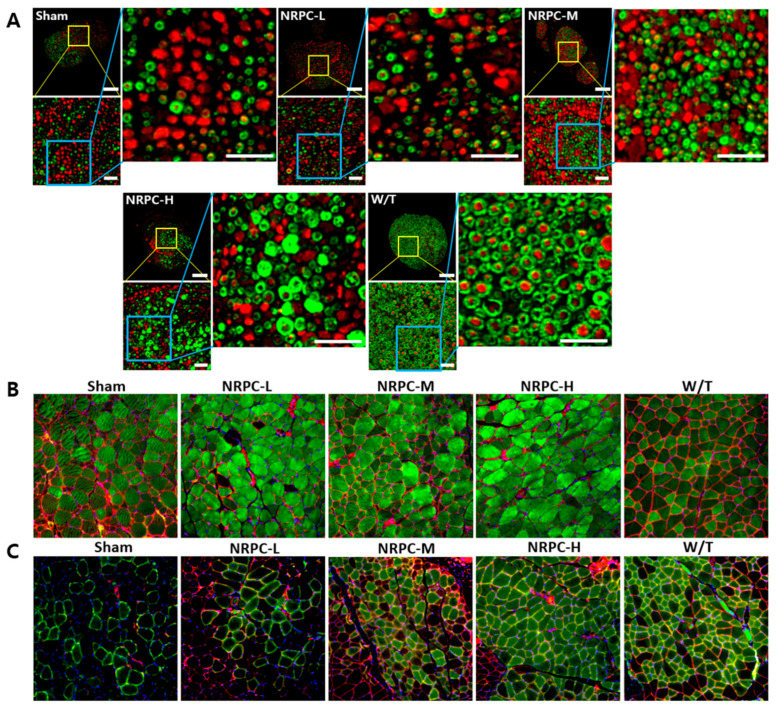
Observation of C22 mouse nerves and muscles with immunohistochemistry in repeated-dose transplantation with NRPCs. (**A**) Axon regeneration and myelin formation of sciatic nerves were confirmed through staining of NF-H (red) and MBP (green). Scale bar of original image: 100 μm. Scale bar of the enlarged yellow and blue box: 20 μm. To confirm the gastrocnemius muscle reconstruction, MYH8 (green) (**B**) or MYH1E (green) (**C**) was double stained with laminin (red), respectively (DAPI (blue)). Original magnification: 200×. NRPCs, neuronal regeneration-promoting cells; NF-H, neurofilament heavy; MBP, myelin basic protein; MYH8, myosin heavy chain 8; MYH1E, myosin heavy chain 1E; NRPC-L, NRPC-low; NRPC-M, NRPC-medium; NRPC-H, NRPC-high; W/T, wild-type.

**Figure 7 biomedicines-11-03334-f007:**
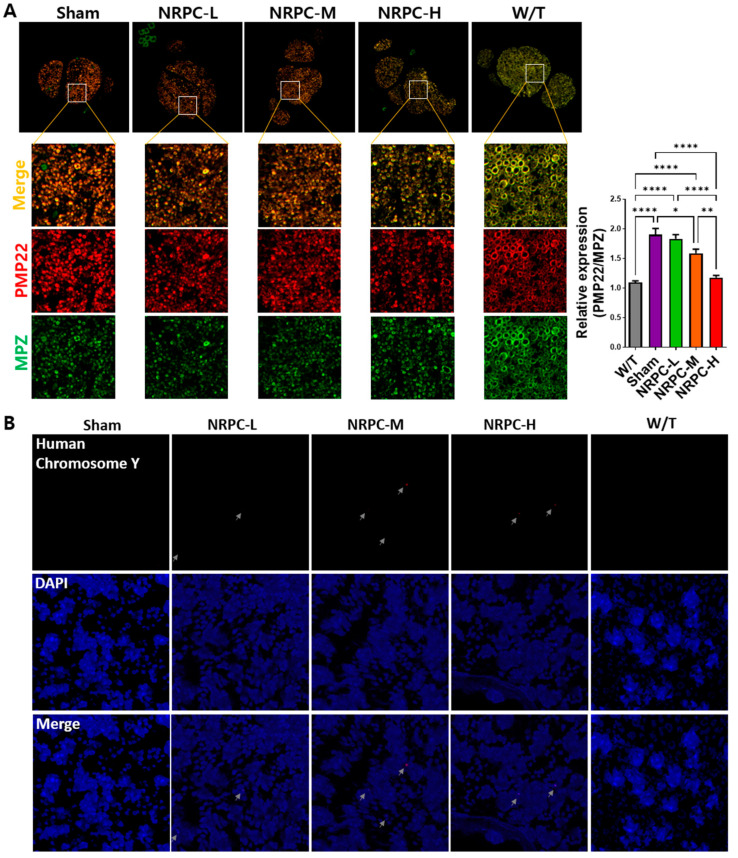
Measurement of PMP22 expression and distribution of NRPCs in the nerves of C22 mice 12 weeks after repeated-dose transplantation. (**A**) NRPCs reduced the expression of PMP22 in the sciatic nerve of C22 mice, which was analyzed using immunostaining (PMP22, red; MPZ, green; merged images, orange). Original magnification: 200×. Protein expression was quantified using ImageJ 1.49 software. PMP22 expression level was normalized to that of MPZ and compared relative to that of the W/T group. Statistical analysis with one-way ANOVA was performed for comparison between groups, and data are presented as mean ± SEM (* *p* < 0.05; ** *p* < 0.01; **** *p* < 0.0001). (**B**) NRPC signals (red) were found in the sciatic nerve of C22 mice using FISH. The red signal pointed at by the gray arrow is the human chromosome Y. Original magnification: 600×. PMP22, peripheral myelin protein 22; NRPCs, neuronal regeneration-promoting cells; MPZ, myelin protein zero; FISH, fluorescence in situ hybridization; NRPC-L, NRPC-low; NRPC-M, NRPC-medium; NRPC-H, NRPC-high; W/T, wild-type.

**Figure 8 biomedicines-11-03334-f008:**
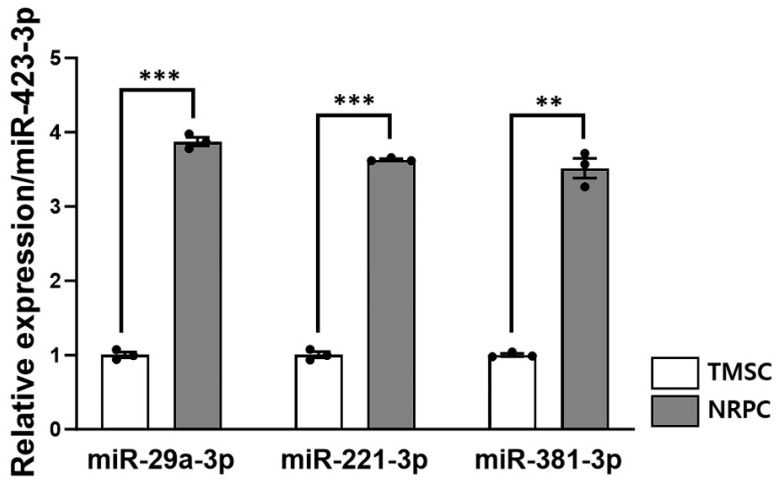
The peripheral nerve regeneration potential of NRPCs. The expression levels of *miR-29a-3p*, *miR-221-3p*, and *miR-381-3p* in TMSC and NRPC were measured using real-time qPCR and normalized to that of *miR-423-3p*. Expression of the three microRNAs was approximately 4fold increased in all NRPCs compared to TMSCs. Statistical comparisons were analyzed using Student’s *t*-tests to compare TMSCs and NRPCs (** *p* < 0.01; *** *p* < 0.001). TMSCs, tonsil-derived mesenchymal stem cells; NRPCs, neuronal regeneration-promoting cells; qPCR, quantitative PCR.

**Table 1 biomedicines-11-03334-t001:** Characteristics of the NRPCs transplanted into C22 mice for preclinical test.

Characteristics	Value/Specification
Viability	>90%
Morphology	Specific form for neural precursor cell of the shape of a bipolar or tripolar
Positive markers	CD73, CD90, CD105
Negative markers	CD14, CD34, CD45
Infectious agent test	SCB (p4), WCB (p9), F/P (p14)
Other quality tests	HIV RNA, HCV RNA, HBV DNA, CMV RNA, anti-HIV, anti-HCV, anti-HBsAg, anti-CMV IgM/IgG, anti-HTLV-1/2, Syphilis
Cryoprotectant	Cryostor CS10

NRPCs, neuronal regeneration-promoting cells; CMV, cytomegalovirus; HBc, hepatitis B core antigen; HBsAg, hepatitis B surface antigen; HBV, hepatitis B virus; HCV, hepatitis C virus; HTLV, human T-cell lymphotropic virus; p, passage; SCB, seed cell bank; WCB, working cell bank; F/P, final product.

**Table 2 biomedicines-11-03334-t002:** Experimental groups of C22 mice used in NRPC single and repeated administration tests.

Experiment	Group	Number of Cells for Transplantation	Mouse	Number of Animals (Male)	Route of Administration	Dosing Frequency
Single administration	NRPC-low	2.5 × 10^4^ cells/ 100 μL/site × 2 sites	C22 mice	11	Intramuscular administration Thigh muscle One site per unilateral leg, administered to both legs	Single
	NRPC-med	2.5 × 10^5^ cells/ 100 μL/site × 2 sites		11		
	NRPC-high	5.0 × 10^5^ cells/ 100 μL/site × 2 sites		12		
	Sham	CS10/100 μL/ site × 2 sites		11		
	W/T	-	Wild-type mice	12		
Repeated administration	NRPC-low	2.5 × 10^4^ cells/ 100 μL/site × 2 sites	C22 mice	11	Intramuscular administration Thigh muscle One site per unilateral leg, administered to both legs	Two repetitions 4 weeks apart
	NRPC-med	2.5 × 10^5^ cells/ 100 μL/site × 2 sites		11		
	NRPC-high	5.0 × 10^5^ cells/ 100 μL/site × 2 sites		11		
	Sham	CS10/100 μL/ site × 2 sites		11		
	W/T	-	Wild-type mice	13		

NRPCs, neuronal regeneration-promoting cells.

## Data Availability

The data that support the findings of this study are available from the corresponding author upon reasonable request.

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
