# Peer review of "Preclinical Efficacy of Peripheral Nerve Regeneration by Schwann Cell-like Cells Differentiated from Human Tonsil-Derived Mesenchymal Stem Cells in C22 Mice"

_biomedicines, 2023, doi:10.3390/biomedicines11123334_

Round 1

Reviewer 1 Report

Comments and Suggestions for Authors

The author's hypothesis is centered around the Preclinical efficacy of peripheral nerve regeneration by Schwann cell-like cells differentiated from human tonsil-derived mesenchymal stem cells in C22 mice, and confirmed the therapeutic effect and peripheral nerve regeneration by transplanting TMSC (TMSC-SC) named neuronal regeneration-promoting cells (NRPC) into C22 mice, a CMT1A disease model. The authors have performed various experiments to prove that neuronal regeneration-promoting cells can be useful for the patients Charcot–Marie–Tooth disease type 1A. The manuscript is well written and nicely presented. However, I have several minor comments that need to be addressed.
1.     Abstract: I will suggest rewriting the conclusion section in a better way.
2.     I will suggest to elaborate introduction section.
3.     How much total RNA was used for cDNA synthesis must be incorporated into the manuscript.
4.     The abbreviation throughout the manuscript should be checked.
5.     Minor grammatical errors should be corrected.
6.     I will suggest adding a graphical abstract, which is the most important point of attraction for a reader.

Author Response

Reviewer: 1

  1. Abstract: I will suggest rewriting the conclusion section in a better way.

Response: According to the reviewer's comment, we revised conclusion section in abstract.

To make clarify, we revised our manuscript as following,

In abstract,

Abstract: Charcot–Marie–Tooth disease (CMT) is a hereditary disease with heterogeneous phenotypes and genetic causes. CMT type 1A (CMT1A) is a type of disease affecting the peripheral nerves and is caused by the duplication of the peripheral myelin protein 22 (PMP22) gene. Human tonsil-derived mesenchymal stem cells (TMSCs) are useful for stem cell therapy in various diseases and can be differentiated into Schwann cell-like cells (TMSC-SCs). We investigated the potential of TMSC-SCs called neuronal regeneration-promoting cells (NRPCs) for peripheral nerve and muscle regeneration in C22 mice, a model for CMT1A. We transplanted NRPCs manufactured in a good manufacturing practice facility into the bilateral thigh muscles of C22 mice and performed behavior and nerve conduction tests and histological and ultrastructural analyses. Significantly, the motor function was much improved, the ratio of myelinated axons was increased, and the G-ratio was reduced by the transplantation of NRPCs. Sciatic nerve and gastrocnemius muscle regeneration of C22 mice following the transplantation of NRPCs downregulated pmp22 overexpression, which was observed in a dose-dependent manner. These results suggest that NRPCs are feasible for clinical research for the treatment of CMT1A patients. Research applying NRPCs to other peripheral nerve diseases is also needed.

  1. I will suggest to elaborate introduction section.

Response: According to the reviewer's comment, we revised the 1st, 2nd and 3rd paragraphs of the introduction to elaborate.

To make clarify, we revised our manuscript as following,

1st, 2nd, and 3rd paragraphs of introduction,

Charcot–Marie–Tooth (CMT) disease is one of the most common hereditary peripheral neuropathies. It is a rare disease with a worldwide prevalence of approximately 1 in 2,500. The muscles in patients with CMT usually atrophy, and the shape of the hands and feet changes abnormally [1,2]. Over 40 genes have been validated in CMT, each of which is associated with one or more disease types. Additionally, one or multiple genes may be linked to one type of CMT [3]. CMT is classified as CMT type 1 (CMT1A), in which the myelin is damaged by genetic mutation; CMT type 2, in which the axon is damaged; CMT type X, with moderate nerve conduction; and CMT type 4, which is inherited as an autosomal recessive trait with impaired myelination [1-3]. The type that was previously named CMT type 3 is now classified as an infantile type (Dejerine–Sottas) or congenital hypomyelinated neuropathy [4].

The most common form, CMT1A, accounts for 60%–70% of total CMT [8]. Patients with CMT1A are characterized by reduced motor nerve conduction velocity (MNCV) <38 m/s, reduced muscle stretches reflexes, and peripheral nerves with an “onion bulb” appearance. The disease occurs via the replication of a 1.4 Mb segment containing the gene for peripheral myelin protein 22 (PMP22) on chromosomes 17p11.2-p12 [5]. PMP22 is mainly expressed in Schwann cells (SCs) constituting the myelin surrounding the axon of the peripheral nerves, and PMP22 is related to myelination during the development of the sciatic nerve. PMP22 is a quadrant helical integral membrane protein that is highly expressed as 2–5% of myelin protein weight in myelin-forming Schwann cells. The properly regulated expression and folding of PMP22 are essential for the development and maintenance of normal myelin in Schwann cells, and when this condition occurs abnormally, the overproduction of pmp22 protein results in CMT1A [6]. Previously, the only methods for treatment were physical rehabilitation, pain control, and walking assistance, but now more than 100 related genes have been discovered, and the development of drug therapies [1, 7–9] and gene therapies [2,10–13] is actively underway.

Mesenchymal stem cells (MSCs) have advantages as therapeutic agents, such as in proliferation, multipotency, immune regulation, and tissue regeneration. Due to continuous research on these MSCs, specific guidelines and quality control methods have been developed, and numerous clinical applications of MSCs are being attempted [14,15]. Compared to MSCs of other origins, tonsil-derived MSCs (TMSCs) have a relatively high yield and proliferation (i.e., shorter doubling times), so they are excellent for quantitative acquisition as a therapeutic agent. Tonsil tissue is readily obtained from a tonsillectomy, the most frequently performed minimally invasive surgery for patients aged 5 to 19 years [15,16]. TMSCs have the potential to differentiate into mesodermal lineages (bone, cartilage, fat, muscle, and tendon), endoderm (hepatocytes, PTH/insulin-releasing cells), and ectodermal lineages (neuron-like cell and glial or Schwann cell-like cells) [15-20].

  1. How much total RNA was used for cDNA synthesis must be incorporated into the manuscript.

Response:

The amount of RNA used for cDNA synthesis was 10 ng.

To make clarify, we added our manuscript as following,

In methods, (page 4, line 185)

MicroRNA isolation was carried out for cells using a mirVana miRNA Isolation Kit (AM1560, Invitrogen, Carlsbad, CA, USA). The isolated total RNA (10 ng) was reverse-transcribed to cDNA using the TaqMan Advanced miRNA cDNA Synthesis Kit (A28007, Applied Biosystems, CA, USA) according to the manufacturer’s instructions.

  1. The abbreviation throughout the manuscript should be checked.

Response:

We revised our manuscript as following,

In methods,

2.1. Cultivation of TMSCs and NRPCs

NRPCs were differentiated from TMSCs as previously described [20].

2.2.5. Nerve conduction study

During the NCS, mice were anaesthetized with a 3% isoflurane (Hana Pharm, Seoul, Korea) in air mixture on a 37°C heating plate. The NCS was performed using Nicolet VikingQuest apparatus (Natus Medical, San Carlos, CA, USA), and compound muscle action potential (CMAP) amplitudes and MNCV were determined. The fur on the distal back and hind limbs was completely removed.

Table 1.

NRPCs, neuronal regeneration-promoting cells; CMV, cytomegalovirus; HBc, hepatitis B core antigen; HBsAg, hepatitis B surface antigen; HBV, hepatitis B virus; HCV, hepatitis C virus; HTLV, human T-cell lymphotropic virus; p, passage; SCB, seed cell bank; WCB, working cell bank; F/P, final product.

Table 2.

NRPCs, neuronal regeneration-promoting cells.

In results,

3.3. Regeneration of the sciatic nerve and gastrocnemius and the related pathway in a single-dose transplantation experiment

Expression of myosin heavy chain 8 (MYH8), the most abundant protein in perinatal skeletal muscle, and laminin, a base substance that provides an anchor for the axons to skeletal muscle, were compared for each experiment group. The expression of MYH8 and laminin was increased in the NRPC groups, in which the staining pattern was similar to that in the W/T mice, and was the highest in the NRPC-high-dose group (Figure 3B). Moreover, the expression of myosin heavy chain 1E (MYH1E), the most abundant protein in adult skeletal muscle, was examined along with laminin.

Figure 1.

Improvement of motor function following single-dose transplantation with NRPCs using gait test and NCS. Image of footprint (A) and graph of SFI (B) to confirm motor function recovery in gate test at 4, 8, and 12 weeks after transplantation. Representative graph of NCS (C) and CMAP and NCV measurements (D) at 12 weeks after transplantation. Statistical analysis with one-way ANOVA was performed for comparison between groups, and data are presented as mean ± SEM. (*p < 0.05; **p < 0.01; ***p < 0.001). NRPCs, neuronal regeneration-promoting cells; NCS, nerve conduction study; SFI, sciatic functional index; CMAP, compound muscle action potential; NCV, nerve conduction velocity; NRPC-L, NRPC-low; NRPC-M, NRPC-medium; NRPC-H, NRPC-high; W/T, wild-type.

Figure 2.

Myelination of the sciatic nerve in C22 mice single-dose transplanted with NRPCs using transmission electron microscope. Representative images at 3000x (A) and 5000x (B) magnification of sciatic nerve cross-sections were presented. Analysis of myelin thickness recovery (C) and ratio of myelinated nerve fibers (D) of the sciatic nerve after NRPC single-dose transplantation. Statistical analysis with one-way ANOVA was performed for comparison between groups, and data are presented as mean ± SEM. (*p < 0.05; **p < 0.01; ***p < 0.001; ****p < 0.0001). NRPCs, neuronal regeneration-promoting cells; NRPC-L, NRPC-low; NRPC-M, NRPC-medium; NRPC-H, NRPC-high; W/T, wild-type.

Figure 3.

Observation of C22 mice nerves and muscles by immunohistochemistry in single-dose transplantation with NRPCs. (A) Axon regeneration and myelin formation of sciatic nerves were confirmed by staining of NF-H (red) and MBP (green). Scale bar of original image: 100 μm. Scale bar of the enlarged yellow and blue box: 20 μm. (B, C) To confirm the gastrocnemius muscle re-construction, MYH8 (green) (B) or MYH1E (green) (C) was double stained with laminin (red), respectively. DAPI (blue). Original magnification 200x. NRPCs, neuronal regeneration-promoting cells; NF-H, neurofilament heavy; MBP, myelin basic protein; MYH8, myosin heavy chain 8; MYH1E, myosin heavy chain 1E; NRPC-L, NRPC-low; NRPC-M, NRPC-medium; NRPC-H, NRPC-high; W/T, wild-type.

Figure 4.

Improvement of motor function following repeated-dose transplantation with NRPCs using gait test, NCS and Rotarod test. Image of footprint (A) and graph of SFI (B) to confirm motor function recovery in gate test at 4, 8, and 12 weeks after transplantation. Representative graph of NCS (C) and CMAP and NCV measurements (D) at 12 weeks after transplantation. Confirmation of motor function recovery in C22 mice repeated-dose transplantation with NRPC by Rotarod test (E). Statistical analysis with one-way ANOVA was performed for comparison between groups, and data are presented as mean ± SEM. (*p < 0.05; **p < 0.01; ***p < 0.001; ****p < 0.0001). NRPCs, neuronal regeneration-promoting cells; NCS, nerve conduction study; SFI, sciatic functional index; CMAP, compound muscle action potential; NCV, nerve conduction velocity; NRPC-L, NRPC-low; NRPC-M, NRPC-medium; NRPC-H, NRPC-high; W/T, wild-type.

Figure 5.

Myelination of the sciatic nerve in C22 mice repeated-dose transplanted with NRPCs using transmission electron microscope. Representative images at 3000x (A) and 5000x (B) magnification of sciatic nerve cross-sections were presented. Analysis of myelin thickness recovery (C) and ratio of myelinated nerve fibers (D) of the sciatic nerve after NRPC repeated-dose trans-plantation. Analysis of the ratio of onion bulb formation (E), the distribution according to diameter of myelinated axons (F), and the correlation between axon diameter and G-ratio (G) by NRPC repeated-dose transplantation in the sciatic nerve. Statistical analysis with one-way ANOVA was performed for comparison between groups, and data are presented as mean ± SEM. (*p < 0.05; **p < 0.01; ***p < 0.001; ****p < 0.0001). NRPCs, neuronal regeneration-promoting cells; NRPC-L, NRPC-low; NRPC-M, NRPC-medium; NRPC-H, NRPC-high; W/T, wild-type.

Figure 6.

Observation of C22 mice nerves and muscles by immunohistochemistry in repeated-dose transplantation with NRPCs. (A) Axon regeneration and myelin formation of sciatic nerves were confirmed by staining of NF-H (red) and MBP (green). Scale bar of original image: 100 μm. Scale bar of the enlarged yellow and blue box: 20 μm. To confirm the gastrocnemius muscle reconstruction, MYH8 (green) (B) or MYH1E (green) (C) was double stained with laminin (red), respectively. DAPI (blue). Original magnification 200x. NRPCs, neuronal regeneration-promoting cells; NF-H, neurofilament heavy; MBP, myelin basic protein; MYH8, myosin heavy chain 8; MYH1E, myosin heavy chain 1E; NRPC-L, NRPC-low; NRPC-M, NRPC-medium; NRPC-H, NRPC-high; W/T, wild-type.

Figure 7.

Measurement of PMP22 expression and distribution of NRPCs in the nerves of C22 mice at 12 weeks after repeated-dose transplantation. (A) NRPCs reduced the expression of PMP22 in the sciatic nerve of C22 mice, which was analyzed using immunostaining (PMP22, red; MPZ, green; merged images, orange). Original magnification 200x. Protein expression was quantified using ImageJ software. PMP22 expression level was normalized to that of MPZ and compared relative to that of the W/T group. Statistical analysis with one-way ANOVA was performed for comparison between groups, and data are presented as mean ± SEM. (*p < 0.05; **p < 0.01; ****p < 0.0001). (B) NRPC signals (red) were found in the sciatic nerve of C22 mice by FISH. The red signal pointed at by the gray arrow is the human chromosome Y. Original magnification 600x. PMP22, peripheral myelin protein 22; NRPCs, neuronal regeneration-promoting cells; MPZ, myelin protein zero; FISH, fluorescence in situ hybridization; NRPC-L, NRPC-low; NRPC-M, NRPC-medium; NRPC-H, NRPC-high; W/T, wild-type.

  1. Minor grammatical errors should be corrected.

Response:

In agreement with your opinion, we have performed English corrections.

  1. I will suggest adding a graphical abstract, which is the most important point of attraction for a reader.

Response:

According to the reviewer's comment, we uploaded the graphical abstract.

Caption of graphic abstract

Schematic illustration of peripheral nerve regeneration after the neuronal regeneration-promoting cells (NRPCs) transplantation into C22 mice. Schwann cell-like cells, NRPCs, were differentiated and produces from human tonsil-derived mesenchymal stem cells. After transplantation of NRPCs, improvement of the ratio of myelinated axons to total axons, downregulation of Pmp22 overexpression and regeneration of the sciatic nerve were observed in C22 mice.

Thank you very much for your valuable comments.

Reviewer 2 Report

Comments and Suggestions for Authors

The authors investigated the potential of TMSC SCs for peripheral nerve and muscle regeneration in C22 mice, a model for CMT1A. They transplanted NRPCs into the bilateral thigh muscles of C22 mice and performed behavior and nerve conduction tests, and histological and ultrastructural analyses. The motor function was improved, the ratio of myelinated axons was increased, and G-ratio was reduced by transplantation of NRPCs. Sciatic nerve and gastrocnemius muscle regeneration of C22 mice following transplantation of NRPCs downregulated pmp22 overexpression. The sciatic nerve regeneration was induced by remyelination of peripheral nerves by NRPC transplantation.

Nam et al. provided substantial data suggesting that NRPCs are useful for the therapy of CMT patients in general.

There is no sufficient data to support the notion that TMSCs are mesenchymal stem cells. The authors are suggested to provide evidence to document the multipotency of these so-called TMSCs. Lacking the data, these cells probably should be referred to as tonsil-derived stem cells and abbreviated TDSCs.

Similarly, there appears no need to coin a new term NRPCs for neuronal regeneration-promoting cells.

Lastly, the writing style is choppy. The English grammar used here is at times difficult to understand. Please work on this.

Comments on the Quality of English Language

The writing style is choppy. The English grammar used here is at times difficult to understand. Please work on this.

Author Response

Author Response to Reviewers’ Comments

Reviewer: 2

Nam et al. provided substantial data suggesting that NRPCs are useful for the therapy of CMT patients in general.

There is no sufficient data to support the notion that TMSCs are mesenchymal stem cells. The authors are suggested to provide evidence to document the multipotency of these so-called TMSCs. Lacking the data, these cells probably should be referred to as tonsil-derived stem cells and abbreviated TDSCs.

Response:

According to reviewer's comment, we provide characteristics of MSCs observed in TMSCs (Figure S2). The characteristics of these cells, such as differentiation capacity and expression of cell surface antigens were met with International Society for Cellular Therapy (ISCT) criteria for mesenchymal stem cells. Therefore, we named these cells as tonsil-derived mesenchymal stem cells from the first publications (Ryu, et al, Cytotherapy, 2012) with these cells. Please understand we called these cells as TMSCs for quite long time.

According to the reviewer’s comment, we revised our manuscript as following,

In results, (page 6)

  1. Results

3.1. NRPCs improved behavioral test results in a single-dose transplantation experiment

The NRPCs used for transplantation into C22 mice were differentiated from TMSCs. The MSC characteristics of TMSCs met the MSC criteria, such as differentiation potential, cell surface antigen, and cell shape, presented by the International Society for Cell Therapy (ISCT) (Figure S2 and S3). Also, the NRPC used in this study exhibited morphological changes after differentiation, such as an elongated bipolar or tripolar spindle shape, with thinner and extended cytoplasm and larger nuclei than TMSCs, as observed in previous studies (Figure S3) [20]. 

We added new Supplementary Figure 2.

Figure S2. Characterization of human TMSCs as MSCs. (A) The profiles of MSC specific surface markers of TMSCs were analyzed by flow cytometry using CD14, CD34, CD45, CD90, CD73, and CD105. The blue histogram profiles indicate the isotype control, and the red histograms indicate a specific antibody. Establishment of the differentiation potential of TMSCs toward mesodermal lineages. TMSCs readily differentiated into osteoblasts, adipocytes, and chondrocytes, as stained by Alizarin Red (B), Oil Red (C), Safranin (D). MSCs, mesenchymal stem cells; TMSCs, tonsil-derived MSCs

We added new Supplementary Figure 3.

Figure S3. Differentiation of NRPCs derived from TMSCs. (A, C) TMSCs were observed under an inverted microscope (x40, x100). (B, D) NRPCs were observed under an inverted microscope (x40, x100). NRPCs exhibited morphological changes after differentiation, such as an elongated bipolar or tripolar spindle shape, with thinner cytoplasmic extensions and larger nuclei than TMSCs. TMSC, tonsil-derived MSC; NRPC, neuronal regeneration-promoting cell.

Similarly, there appears no need to coin a new term NRPCs for neuronal regeneration-promoting cells.

Response:

We have already reported on a paper of Schwann-like cells differentiated from tonsil-derived mesenchymal stem cells (TMSC-SCs). The NRPCs were named after the characteristics of TMSC-SCs as Schwann cells and the characteristics of the neurotrophic effect of these cells in our previous studies. These Schwann cell-like cells exhibited the characteristics of Schwann cells, such as myelination and also exhibited enhance neuronal outgrowth from neuronal cells, such as NSC34 cells (Jung, et al. Int J Mol Sci, 2016). Therefore, reflecting the characteristics of cells, the cells were referred to NRPCs. We already described the characteristics of these cells in discussion (page 16, lines 494-504). Please understand we called these cells as NRPCs in this experiment.

According to the reviewer’s comment, we would like to clarify this comment as following,

We revised and added new sentences in Introduction,

In introduction (page 2),

In the present study, we confirmed the therapeutic effect and peripheral nerve regeneration caused by transplanting TMSC-SCs named neuronal regeneration-promoting cells (NRPC) into C22 mice, a CMT1A disease model. The NRPCs were named after the characteristics of TMSC-SCs as Schwann cells and the characteristics of the neurotrophic effect of these cells in a previous study [25].

Lastly, the writing style is choppy. The English grammar used here is at times difficult to understand. Please work on this.

Response:

In agreement with your opinion, we have received English editing service after careful correction by ourselves.

Thank you very much for your valuable comments.

Round 2

Reviewer 2 Report

Comments and Suggestions for Authors

The NRPCs used for transplantation into C22 mice were differentiated from TMSCs. The MSC characteristics of TMSCs met the MSC criteria, such as differentiation potential, cell surface antigen, and cell shape, as provided in the supplemental figures (Figures S2 and S3).

The manuscript has been revised as suggested.